# Generating a Novel Dataset for Mechanisms of Drug-Induced Toxicity using LLM-supported tools

**Sara Masarone**
Ignota Labs
Cowley Rd, Milton, Cambridge CB4 0WS
sara.masarone@ignotalabs.ai

**Alex Beeson**
University of Warwick
Coventry, CV4 7AL
alex.beeson@warwick.ac.uk

**Katie V. Beckwith**
Ignota Labs
Cowley Rd, Milton, Cambridge CB4 0WS
katie.beckwith@ignotalabs.ai

**Matthew Mason**
Ignota Labs
Cowley Rd, Milton, Cambridge CB4 0WS
matthew.mason@ignotalabs.ai

**Thomas Clelford**
Ignota Labs
Cowley Rd, Milton, Cambridge CB4 0WS
thomas.clelford@ignotalabs.ai

**Arran Willmott**
Ignota Labs
Cowley Rd, Milton, Cambridge CB4 0WS
arran.willmott@ignotalabs.ai

**Giovanni Montana**
Warwick University
Coventry, CV4 7AL
g.montana@warwick.ac.uk

**Layla Hosseini-Gerami**
Ignota Labs
Cowley Rd, Milton, Cambridge CB4 0WS
layla.gerami@ignotalabs.ai

## Abstract

Toxicity is a leading cause of drug failure, yet existing resources often lack the mechanistic context linking drug perturbations to adverse outcomes. To bridge this gap, we introduce ToxMech, an ongoing project developing an LLM-supported system that extracts and structures toxicity mechanisms into a comprehensive heterogeneous knowledge graph.

ToxMech integrates data from diverse sources, including PubMed, AOP-Wiki, FDA boxed warnings, and clinical news, using retrieval-augmented agents to mine both structured repositories and unstructured text. By encoding relationships between drugs, targets, pathways, and outcomes, ToxMech enables structured reasoning over the causal chains of drug-induced toxicity. This evolving resource aims to provide researchers with a robust tool for mechanistic modelling and enhanced safety assessment in the drug discovery pipeline.

## 1 Introduction

Drug toxicity is a primary cause of late-stage clinical failure, with up to 30% of candidates failing due to unacceptable safety profiles Tran et al. (2023); Cook et al. (2014). These failures result in significant financial loss and delayed patient access to novel treatments Peter Guengerich (2010). Improving the ability to anticipate and minimise toxicity is therefore a central challenge in computational drug discovery.

A major barrier to progress is the lack of structured, mechanistically rich datasets. Existing cheminformatic models often lack access to the curated data necessary to generalise beyond specific chemical scaffolds or assay conditions Walters & Barzilay (2021); Masood et al. (2025). A systematic repository of mechanistic toxicity data would enable more accurate predictive modelling and provide the interpretability required to redesign safer compounds.

To address this need, we are developing ToxMech, an ongoing project to create a benchmark dataset that systematically describes the mechanisms of drug toxicity. By providing a shared resource for the research community, ToxMech aims to facilitate reproducibility, improve model evaluation, and accelerate the discovery of safer therapeutics in high-stakes clinical domains.

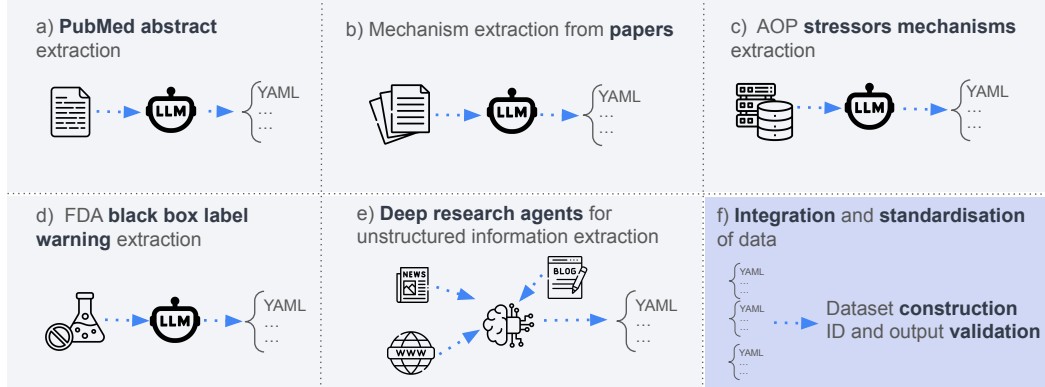

Figure 1: The figure illustrates a multi-step pipeline for extracting, integrating and standardising biomedical information using LLMs. PubMed abstracts (a), mechanisms from papers (b), AOP stressor mechanisms (c), FDA black box warnings (d), and unstructured sources (ie. blogs, news) (e) are processed into YAML outputs, which are then integrated and validated for dataset construction (f).

## 2 LANDSCAPE AND LIMITATIONS

Existing toxicity resources remain fragmented and lack the mechanistically rich, unified structure required for causal machine learning. Although the AOP framework Villeneuve et al. (2014) maps causal chains from molecular events to outcomes, its curation is heterogeneous and lacks a therapeutic drug focus. Similarly, DrugMechDB Gonzalez-Cavazos et al. (2023) provides structured mechanisms but prioritises efficacy over failure modes. Clinical resources such as SIDER, OFF-SIDES, and OnSIDES Chandak & Tatonetti (2020); Kuhn et al. (2016); Tatonetti et al. (2012) provide standardised drug-to-outcome mappings but fail to explain the underlying biological reasons for these associations. High-throughput platforms and organ-specific datasets, including Tox21, Tox-Cast, ToxicoDB, DILIrank, CiPA, and MitoTox Richard et al. (2016); Nair et al. (2020); Chen et al. (2016); Konala et al. (2025); Lin et al. (2021), offer extensive signalling and assay data. However, these resources rarely bridge the gap between assay-level perturbations and clinical observations. Finally, while MoleculeNet, Therapeutics Data Commons Huang et al. (2022); Wu et al. (2017), and UniTox Silberg et al. (2024) have successfully standardised these subsets for benchmarking, they remain focused on predictive classification rather than explicit mechanistic reasoning. Collectively, these resources represent a landscape where progress is limited by a lack of causal integration.

## 3 DATASET GENERATION

To facilitate a mechanistic understanding of toxicity, we are developing ToxMech, a hybrid LLM-supported framework that generates a standardised dataset of drug-induced toxicity mechanisms. Our pipeline integrates multi-modal data into structured YAML format, enabling the construction of high-fidelity graphs for computational analysis. Early results demonstrate that this approach successfully bridges the gap between molecular initiating events and clinical outcomes. The following sections detail how our modules currently populate the ToxMech dataset, addressing a landscape of progress halted by lack of causal integration.

## 3.1 PUBMED LITERATURE MINING

The first modality leverages PubMed as a primary source of mechanistic knowledge. To construct a validation dataset for drug-induced toxicity, we developed an automated scraping pipeline using Entrez Programming Utilities (E-utilities). The module systematically retrieves literature by executing targeted queries to identify relevant PubMed identifiers (PMIDs). For each PMID, the pipeline downloads XML records and parses them to extract structured metadata, specifically article titles and abstracts. This information is aggregated into a machine-readable dictionary format for subsequent analysis.

This structured dataset is then processed by GPT-4o to extract mechanistic insights, such as sequences of biological events, which are mapped to predefined node types within our knowledge representation framework. The final output is serialised as a YAML file. Figure 1 summarises this construction process. Details o the prompt used for the YAML file can be found in the appendix A.1.

## 3.2 EXTRACTING INFORMATION FROM MECHANISTIC PAPERS

In addition to PubMed retrieval, the pipeline integrates mechanistic insights from full-text scientific papers to capture experimental evidence often omitted from abstracts. We utilise the CORE API to retrieve open-access PDFs, which are preprocessed and converted into a machine-readable format. The text is then segmented and processed by an LLM-supported pipeline to extract structured chains of cause and effect underlying drug-induced toxicity.

## 3.3 USING GPT4O FOR UNSTRUCTURED DATA EXTRACTION

To complement structured literature sources, we used GPT-4o to systematically retrieve and process mechanistic information on drug-induced toxicity from the web. The pipeline begins by obtaining a comprehensive list of drugs from the DrugBank dataset, ensuring coverage of clinically relevant compounds Knox et al. (2024). For each drug in the list, GPT-4o identifies and aggregates mechanistic evidence related to adverse drug reactions. The outputs are constrained to return only a maximum of five causal steps, thus limiting variability and intending to provide a concise explanation of the mechanism of action.

This evidence is synthesised into a structured report that summarises all known mechanisms of drug-induced toxicity for a given compound. The language model then parses the report to extract distinct mechanistic pathways, each of which is converted into a dedicated YAML file formatted according to the requirements of our validation framework. In cases where multiple toxicity mechanisms are identified for a single drug, the system generates multiple YAML files, each representing a unique mechanistic hypothesis. These files are stored individually, enabling downstream integration with other data sources and facilitating validation of predicted mechanisms.

## 3.4 AOP STRESSORS SCRAPER

To incorporate mechanistic insights from established pathways, we developed an automated scraper to link chemical stressors to biological events defined in AOP-Wiki. Building upon a previously extracted dataset of stressors, the pipeline iterates through each entry and its associated AOP identifiers to execute targeted HTTP requests. The scraper parses the Events section of each AOP page, extracting titles and descriptions that detail key biological processes.

These event titles provide the mechanistic context necessary to link chemical exposures to specific molecular or cellular outcomes. The resulting data is stored in a structured JSON format, mapping each stressor to its respective event sequence. This approach enriches the validation dataset with high-quality annotations that align with internationally recognised toxicological knowledge. Figure 2 illustrates this mechanistic structure through the example of hERG channel blockade.

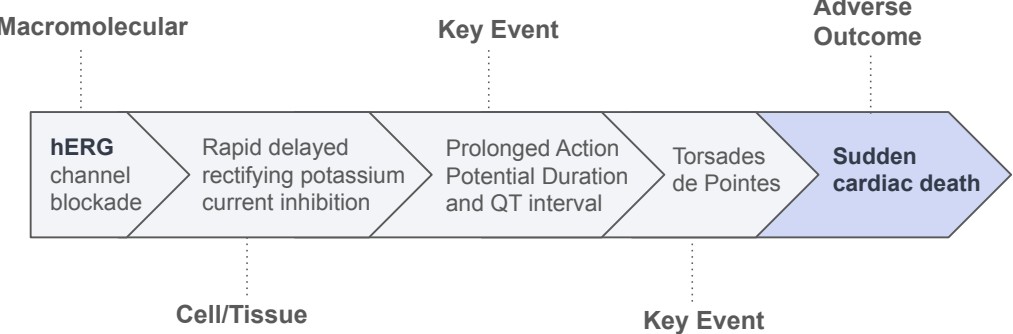

Figure 2: Summary of the "hERG channel blockade" Adverse Outcome Pathway.

## 3.5 FDA SIDE EFFECTS EXTRACTOR

We further supplemented this dataset by extracting boxed warning data from the FDA website, which registers life threatening and serious adverse reactions associated with approved medications in the US Delong & Preuss (2023).

## 3.6 DATA HARMONISATION

Once the data has been gathered, it is harmonised by aligning extracted entities with standardised biological identifiers (ie. Drugbank IDs, UniProt). The yaml file harmonisation adopts the BioLink ontology model, utilised in with many KG frameworks Unni et al. (2022). This approach allows us to leverage the depth of full text articles, providing richer mechanistic context than abstracts alone and significantly enhancing the comprehensiveness of the resulting validation dataset.

## 3.7 STATS AND INFO ON THE DATASET - THIS COULD BE A TABLE

The table below summarises the dataset to date (subject to update as it's a work-in-progress).

Table 1: ToxMech dataset stats

| STATS | N |
|---|---|
| N of drugs | 1534 |
| N of edges | 7670 |
| Average N of nodes per mechanism | 5 nodes plus edges |

## 4 EVALUATION

To ensure the system is sufficiently robust, we employ a multi-tiered validation strategy that leverages both human expertise and automated verification. The first tier involves a manual evaluation performed on a randomised 10% subset of the total dataset to serve as a high-fidelity benchmark. A panel of three experts assesses each mechanism for biological plausibility and directionality (see section A.2 for additional details). Initial results demonstrate a strong inter-rater reliability with a $\kappa = 0.78$. This human-validated subset provides the ground truth necessary to calibrate the subsequent layer of the pipeline. In the second tier, an automated verification agent utilising an independent GPT-4o instance with a temperature of 0.0 performs cross-modal fact-checking. This agent decomposes complex causal chains into individual atomic claims and cross-references them against source metadata to identify logical inconsistencies or hallucinations. By integrating this expert-led validation with granular automated auditing, we establish a scalable framework that pre-

vents progress halted by lack of causal integration and ensures the mechanistic integrity of the final dataset. Examples of two extracted pathway can be found in the appendix A.3.

## 4.1 LIMITATIONS

To address instances where the extracted pathway is found to be inconsistent with its original source material, we utilise a structured expert-led reconciliation protocol. The secondary GPT-4o verification agent is tasked with flagging these specific discrepancies, at which point the affected causal chain is categorised as ambiguous and diverted to a manual adjudication queue. This human-in-the-loop stage is essential for maintaining data fidelity, as it ensures the final dataset is not reliant on automated judgements that might misinterpret complex biological evidence. By focusing expert review on these high-uncertainty cases, we safeguard the mechanistic integrity of the results.

## 5 CONCLUSION

This paper presents early results from a system that aims to address the scarcity of high-quality, mechanistic data for drug-induced toxicity prediction. By systematically constructing a mechanism based framework for drug induced toxicity, we not only introduce a structured method for capturing complex biological interactions, but also aim to generate a rich and informative dataset which we plan to release upon completion.

By informing machine learning with biological processes rather than fragmented data, we enhance the predictive power and interpretability of toxicity assessments. This approach supports safer therapeutic development in a field where research is often constrained by a lack of causal integration.

## 6 MEANINGFULNESS STATEMENT

ToxMech addresses the workshop goal of learning meaningful representations of life by transforming fragmented data into an interpretable, mechanistic knowledge graph. By integrating multimodal data from abstracts to structured omics and pathway repositories, it constructs a multi-scale representation of biological systems under chemical perturbation. Crucially, ToxMech moves beyond black-box predictions by focusing on causal representation learning, mapping the interventional chain from molecular interactions to disease phenotypes. This aligns with themes of modelling biology across scales and creating constraint-aware representations, providing a generalisable framework for researchers to decode the complex, non-linear dynamics of drug-induced toxicity.

ACKNOWLEDGMENTS

The authors wish to acknowledge the entire team for their technical insights and foundational work on the tech tools and orchestrating architecture. This project would not have been possible without the shared dedication and multidisciplinary expertise of the entire laboratory.

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

## A APPENDIX

### A.1 PROMPT

```
You are an expert in drug toxicity mechanisms. Extract mechanistic
    information ONLY about ADVERSE/TOXIC effects.

CRITICAL FILTER: ONLY process papers about DRUG-INDUCED TOXICITY or
    ADVERSE EFFECTS.
REJECT papers about:
   - Therapeutic benefits
   - Positive drug effects
   - Disease treatment mechanisms
   - Drug efficacy

OUTPUT FORMAT:
```
DRUG_NAME: <drug name in lowercase>

YAML_FILE:
```yaml
graph:
  _id: '<DrugBank_ID>'
  drug: '<drug name>'
  toxicity_type: '<specific toxicity type>'
  mechanism_text: '<2-3 sentence plain-English explanation of how
      the drug causes toxicity>'

nodes:
  - id: '<DrugBank ID>'
    label: 'Drug'
    name: '<drug name>'
  - id: '<UniProt or GO ID>'
    label: 'Protein' or 'BiologicalProcess'
    name: '<protein or process name>'
  [... more nodes as needed ...]

links:
  - source: '<node id>'
    target: '<node id>'
    relation: '<relationship from excel file>'
  [... more links as needed ...]
reference:
  - '<paper title or PMID>'
CRITICAL RULES:
1.      graph._id format: DrugBankID (e.g., 'DB00001')
2.      graph.drug: lowercase drug name
3.      graph.toxicity_type: MUST be specific (hepatotoxicity,
   cardiotoxicity, nephrotoxicity, neurotoxicity, cytotoxicity, etc
   .) - NOT just  toxicity
4.      graph.mechanism_text: Write 2-3 sentences in plain English
   explaining HOW the drug causes toxicity. Include: what the drug
   binds/affects     what pathway is altered     what toxic outcome
    occurs.
5.      nodes: Each must have id, label, name
   - Drug nodes: label='Drug', id=DrugBank ID
   - Protein nodes: label='Protein', id=UniProt ID
   - Process nodes: label='BiologicalProcess', id=GO ID
6.      links: Must use relationships from this list: {content3}
7.      NEVER use MESH, UBERON, or DBXXXXX placeholders
8.      Include paper title in reference
9.      Show mechanism flow: Drug      Protein/Enzyme      Pathway/
   Process      Toxicity Outcome
```

## A.2    EVALUATION DETAILS

The human-grounded evaluation was conducted on a stratified randomised subset comprising 10 per cent of the total dataset. This assessment was performed by three independent experts, each holding a PhD in molecular biology, bioinformatics or cheminformatics with a minimum of five years of experience in this field. The experts evaluated each mechanism against two primary criteria: biological plausibility and the accuracy of causal directionality. Inter-rater reliability was calculated using Cohen's Kappa. The early results indicated substantial agreement and provided a high-fidelity benchmark for the system. By using this expert-led evaluation to calibrate the model, we ensure that the dataset's development is not a case where progress is halted by lack of causal integration.

## A.3    CASE STUDIES

To demonstrate that the constructed mechanisms reflect current scientific understanding, we performed a preliminary evaluation of two representative examples from our broader dataset. By benchmarking these specific cases against peer-reviewed literature on drug-induced toxicity pathways, we aimed to provide an initial validation of the model's accuracy. This pilot assessment serves as a proof of concept, illustrating the alignment of our components with established biological data across the many mechanisms we have evaluated.

## A.4    MECHANISM DESCRIPTION 1: "ABCIXIMAB CAUSES (EXCESSIVE) BLEEDING"

Causal steps identified:

Abciximab (node) → "binds to" (edge) → "GP IIb/IIIa receptor (UniProt:P08514)" (node) → (4) [and] "inhibits" (edge)→ (5) "platelet aggregation". Abciximab (node) → "synergises with" (edge) → "Anticoagulants"

**Original source:** "Abciximab General Information Abciximab is a Fab fragment of the chimeric human-murine monoclonal antibody 7E3, which binds to the platelet glycoprotein IIb/IIIa receptor and inhibits platelet aggregation. Abciximab is used for prevention of cardiac ischemic events in patients undergoing percutaneous coronary intervention and to prevent myocardial infarction in patients with unstable angina who do not respond to conventional treatment. It has also been used for thrombolysis in patients with peripheral arterial occlusive disease and arterial thrombosis. Besides bleeding, other adverse reactions that have been associated with abciximab include back pain, hypotension, nausea, and chest pain (but with an incidence not significantly different from that observed with placebo)." Oldenburg (2019).

## A.5    MECHANISM DESCRIPTION 2: "ACETAMINOPHEN CAUSES HEPATOTOXICITY"

Causal steps identified:

Acetaminophen (node) → "metabolized by" (edge) → "Cytochrome P450 2E1" (node) → (1) [and] "produces" (edge) → (2) "NAPQI" (node) → (3) [and] "depletes" (edge) → (4) "glutathione" (node) → (5) [and] "leads to" (edge) → (6) "oxidative stress" (node) → (7) [and] "causes" (edge) → (8) "hepatocellular damage" (node) → (9) [and] "results in" (edge) → (10) "Hepatotoxicity".

**Original source:** "When taken at therapeutic doses, most of APAP is metabolized by phase II conjugating enzymes, mainly UDP-glucuronosyltransferase (UGT) and sulfotransferase (SULT), converting it to nontoxic compounds which are then excreted with the urine. Only a very small portion is excreted unchanged in the urine. The remaining APAP, approximately 5–9%, is metabolized by the cytochrome P450 enzymes (CYPs), mainly CYP2E1, into the highly reactive intermediate metabolite N-acetyl-p-benzoquinone imine (NAPQI). Generally, NAPQI is rapidly detoxified by conjugating with glutathione (GSH). However, when phase II metabolizing enzymes are saturated after APAP overdose, excessive NAPQI depletes GSH, leading to covalent binding of sulfhydryl groups in cellular proteins, especially mitochondrial proteins. This results in mitochondrial oxidative stress and dysfunction, ultimately causing hepatocyte necrosis." Yan et al. (2018).

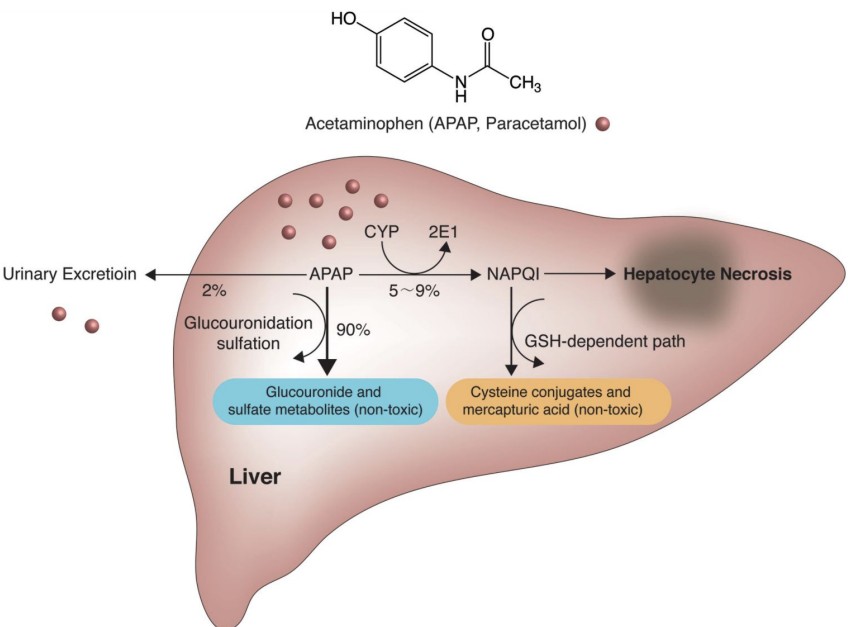

Figure 3: Mechanism of drug induced tonicity with Acetaminophen. At therapeutic doses, most acetaminophen (APAP) is safely metabolized and excreted, with only a small portion forming the reactive metabolite NAPQI that is detoxified by glutathione (GSH). In overdose, detox pathways are overwhelmed, GSH is depleted, and excess NAPQI binds cellular proteins, causing mitochondrial dysfunction and hepatocyte necrosis Yan et al. (2018).

