# OpenReview forum: "Generating a Novel Dataset for Mechanisms of Drug-Induced Toxicity using LLM-supported tools"
_ICLR.cc/2026/Workshop/LMRL — ICLR 2026 Workshop LMRL Poster_

### Official Review · Reviewer_w3fZ · 2026-02-23
**Promising Mechanistic Toxicity KG Construction, but Benchmark Definition Remains Incomplete**

**Rating:** 6
**Confidence:** 3

**Review:**

ToxMech proposes an LLM-assisted pipeline to construct a mechanistic knowledge graph of drug-induced toxicity for causal representation learning.

### Pros

- Encodes explicit causal chains (Drug → Protein → Process → Outcome), aligning well with meaningful biological representations.
- Integrates multi-source data into a BioLink-aligned schema with human validation (κ = 0.78).

### Cons

- Benchmark setup is unclear (no defined tasks, baselines, metrics, or splits).
- Limited quantitative analysis of dataset quality, bias, and LLM error patterns.

### Minor Comments

- Edge type vocabulary is not explicitly listed.
- In A.4, *Abciximab → “synergises with” → Anticoagulants* is hard to interpret as a mechanistic causal step.
- Adding explicit GO IDs in A.4 and A.5 would improve ontology clarity.

---

### Official Review · Reviewer_Bk52 · 2026-02-26
**ToxMech**

**Rating:** 5
**Confidence:** 3

**Review:**

This work presents ToxMech, an LLM-supported pipeline for constructing a mechanistic drug-induced toxicity dataset by integrating multi-source data (PubMed, AOP-Wiki, FDA boxed warnings) into a heterogeneous knowledge graph. The pipeline uses GPT-4o for unstructured data extraction and a multi-tiered validation strategy (human expert + automated fact-checking) to ensure data quality. The work addresses a critical gap in mechanistic toxicity data for drug discovery, with high practical significance, though the dataset is still in progress and some validation details need further refinement.
﻿
Pros
Addresses a critical gap in the field: the lack of structured, mechanistically rich drug-induced toxicity datasets, which limits the development of interpretable toxicity prediction models.
Multi-source data integration (PubMed, AOP-Wiki, FDA) that links molecular events to clinical outcomes, creating a comprehensive knowledge graph for mechanistic toxicology.
﻿
Cons
The dataset is still in progress, with only preliminary stats (1534 drugs) reported – the paper does not specify the final target size, coverage of toxicity types (e.g., cardiotoxicity, hepatotoxicity), or inclusion criteria for drugs/toxicity mechanisms.
The automated fact-checking module is underdescribed – no details on the prompt engineering, metrics for hallucination detection, or false positive/negative rates of the automated validation.

---

### Meta-Review · Area_Chair_DuKA · 2026-02-27

**Recommendation:** Accept (Poster)
**Confidence:** 3

**Metareview:**

Accept.

---

### Decision · Program_Chairs · 2026-03-02

**Decision:**

Accept (Poster)

**Comment:**

Please see the meta-review.